## A highly-efficient automated optimization approach for kilometerlevel resolution Earth system models on heterogeneous many-core supercomputers

Xiaojing Lv<sup>1,2\*</sup>, Zhao Liu<sup>1\*</sup>, Yuxuan Li<sup>5\*</sup>, Shaoqing Zhang<sup>3,4\*</sup>, Haohuan Fu<sup>1,5\*</sup>, Xiaohui Duan<sup>1,6\*</sup>, Shiming Xu<sup>5</sup>, Yang Gao<sup>3,4</sup>, Yujing Fan<sup>1</sup>, Lifeng Yan<sup>6</sup>, Haopeng Huang<sup>1,5</sup>, Haitian Lu<sup>1</sup>, Lingfeng Wan<sup>3,4</sup>, Haoran Lin<sup>6</sup>, Qixin Chang<sup>6</sup>, Chenlin Li<sup>1</sup>, Quanjie He<sup>1</sup>, Yangyang Yu<sup>3</sup>, Qinghui Lin<sup>7</sup>, Sheng Jia<sup>7</sup>, Tengda Zhao<sup>7</sup>, Weiguo Liu<sup>1,6</sup>, Guangwen Yang<sup>1,5</sup>

Abstract. As coupled Earth system models advance, it becomes increasingly feasible to attain higher spatial resolutions, thereby enabling more precise simulations and predictions of the evolution of the Earth system. Consequently, there is an urgent demand of highly-efficient optimization for extensive scientific programs on more power-efficient heterogeneous many-core systems. This study introduces a highly-efficient optimization approach tailored for kilometer-level resolution Earth System Models (ESMs) operating on heterogeneous many-core supercomputers. Leveraging scalable model configurations and innovative tripolar ocean/sea-ice grids that bolster spatial accuracy and computational efficiency, we initially establish a series of high resolutions (HRs) within a solitary component (either the atmosphere or ocean) while maintaining a fixed resolution for the other, resulting in notable enhancements in both model performance and efficacy. Furthermore, we have devised an OpenMP tool specifically optimized for the new Sunway supercomputer, facilitating automated code optimization. Our approach is designed to be non-intrusive, minimizing the need for manual code alterations while ensuring both performance gain and code consistency. We adopt a hybrid parallelization strategy combining Athread and OpenMP, achieving full parallel coverage for code segments with a runtime proportion exceeding 1%. After optimization, the atmosphere, ocean, and sea-ice models achieve speedups of 4.43×, 1.86×, and 2.43×, respectively. Consequently, the overall simulation performance of the 5-km/3-km coupled model reaches 222 SDPD. This achievement renders multiple decadal scientific numerical simulations utilizing such HR coupled simulations feasible. Our work signifies a pivotal advancement in Earth system modeling, providing

<sup>&</sup>lt;sup>1</sup>National Supercomputing Center in Wuxi, Wuxi, China

<sup>&</sup>lt;sup>2</sup>China Ship Scientific Research Center, Wuxi, China

<sup>&</sup>lt;sup>3</sup>Key Laboratory of Physical Oceanography, the College of Oceanic and Atmospheric Sciences & Institute for Advanced Ocean Study, Ocean University of China, Qingdao, China

<sup>&</sup>lt;sup>4</sup>Laboratory for Ocean Dynamics and Climate, Qingdao Center for Marine Science and Technology, Qingdao, China

<sup>&</sup>lt;sup>5</sup>Tsinghua University, Beijing, China

<sup>&</sup>lt;sup>6</sup>School of Software, Shandong University, Jinan, China

<sup>&</sup>lt;sup>7</sup>Qingdao Gosci Technology Group, Qingdao, China

<sup>\*</sup>Co-first authors contribute equally to this work.

<sup>\*</sup>Correspondence to: Shaoqing Zhang (szhang@ouc.edu.cn), Haohuan Fu (<u>haohuan@tsinghua.edu.cn), Xiaohui</u> Duan (sunrise.duan@sdu.edu.cn)

a robust framework for high-resolution climate simulations on more ubiquitous (next-generation) heterogeneous supercomputing platforms, such as GPUs, with minimal additional effort.

**Keywords:** Automated Optimization; OpenMP-based; Kilometer-level resolution Earth system models; Heterogeneous supercomputing.

#### 1 Introduction

Coupled Earth System Models (ESMs) play a crucial role in understanding historical, current, and future climate states with components such as atmosphere, ocean, land, and ice to represent the interactions and feedback processes in the climate system (Randall et al., 2007; Stocker et al., 2013; Yazdandoost et al., 2021). Recently, the development of high-resolution coupled Earth system models and studying the impact of meso- and small-scale interactions on climate simulation and forecasting has demanded on kilometer-level climate modeling with expectation of reducing current climate model uncertaintiess to a new level (Zhang et al., 2014). Examples include the U.S. Energy Exascale Earth System Model (E3SM) (Caldwell et al., 2021) and the China Sunway high-resolution coupled Earth system models (SW-HRESMs) (Zhang et al., 2023). The development of coupled ESMs enables a more accurate simulation of multiscale interactions, and integrating these intricate and precise elements results in more informed and accurate climate projections (Benjamin et al., 2019). While more and more scientific understanding converted into software modules, more and more grids in the model to push the resolution from a few hundred kilometers to a few kilometers, the demand for high-performance computing (HPC) resources and highly-efficient optimization of massive scientific programs on more power-efficient supercomputer systems are eagerly demanded (Zhang et al., 2020).

In recent decade, the architecture of supercomputers has unconsciously diverted and gradually shifted from isomorphic multi-core to heterogeneous multi-core systems as the multi-core supercomputers are limited by Moore's Law (Zhang et al., 2020). Compared with homogeneous multi-core computer systems, heterogeneous systems integrate co-processors or accelerators with different instruction sets and performance characteristics, and the memory hierarchy is changed radically. As pointed out by previous studies (Duan et al., 2024), although substantial advantages are projected for both efficiency and performance of the heterogeneous multi-core system, the architectural change brings tough challenges on many different aspects. These challenges include porting, parallelizing and optimizing of heavy legacy codes from homogeneous to heterogeneous, as well as MPI communication, parallel I/O and reproducibility etc.

High-resolution climate models in heterogeneous supercomputers have begun with migration and optimization dealing with severe challenges from both scientific application demands and new architecture programming demands (Fu et al., 2016; Zhang et al., 2023), based on previous efforts on the Nonhydrostatic Iscosahedral Atmospheric Model (NICAM) (Dennis et al., 2012; Satoh et al., 2008) and the Earth simulator (Fudeyasu et al., 2008) on the K computer. Johnsen et al. (2013) detail high-resolution simulations using the Community Atmospheric Model's spectral element dynamical core (CAM-SE) on the CPU cores of Jaguar. Similarly, the work by Leung et al. (2020) demonstrates the application of the Weather Research and Forecasting (WRF) model on the CPU cores of Blue Waters.


Multiple-year or even longer kilometer-scale climate simulation is on the road of exascale systems coming into concrete plans. For example, with a goal to integrate model development with the leading-edge computational power, E3SM (Energy Exascale Earth System Model) targets ultra-high resolution with comprehensive infrastructure (Miyamoto et al., 2013). However, following efforts upgrade E3SM (Williams, 2016) to develop a model that nearly doubles the speedup speed and with an improved climate simulation in many aspects. This transition significantly enhances the model's speed, nearly double, and enhances the accuracy of climate simulation. Nevertheless, the majority of outcomes have solely been assessed on an AMD CPU platform. The previously optimized KNL architecture is obsolete, and there is still an impending need for thorough model acceleration. HOMMEXX-NH is a highly optimized iteration of E3SM's non-hydrostatic dycore. The system has a capacity to accommodate up to 276,000 GPUs simulating 0.97 years in a single day (Golaz et al., 2022). The porting of HOMMEXX-NH requires code refactoring of its dycore solver using the Kokkos framework (Bertagna et al., 2020). Such revision provides versatile efficiency across different accelerator architectures. Furthermore, it presents additional learning obstacles for the climate Fortran community. Similar porting and optimizing strategy are applied to COSMO achieving a performance of 0.043 SYPD for a near-global 1 km resolution configuration by utilizing 4,888 GPUs of the Piz Daint supercomputer (Fuhrer et al., 2018), with the introduction of Stencil Loop Language (STELLA) (Gysi et al., 2015).

Previous studies made significant improvements in CAM by manually rewriting and substantially fine-tuning substantial quantities of code (Fu et al., 2016; Schneider, 2022). Following work on the CAM, we adopted system-level tuning for communication and load balancing to the other components of the model to complete the very first heterogeneous version of high-resolution CESM (25-km atmosphere and 10-km ocean), i.e. CESM-HR\_sw1.0 (Zhang et al., 2020). CESM-HR\_1.0 has produced a pioneering dataset of long-term, high-resolution simulations for the Earth system (Chang et al., 2020). While previous studies primarily deal with the processing of atmospheric components, the most similar comparison to our work in terms of transferring and enhancing the full-interconnected model is the ICON-Sapphire configuration (Sato et al., 2020) of the ICON Earth System Model (Hohenegger et al., 2023). Since 2015, the research team at the Max Planck Institute for Meteorology (MPI-M) has been working on adapting the ICON model to the GPU using OpenACC instructions (Schär et al., 2020; Jungclaus et al., 2022), with the aim of achieving their Earth system model objectives. This effort was initiated after the development of the non-hydrostatic dynamic solver. The model is specifically built to depict global climate phenomena at a resolution of several kilometers, offering an unparalleled level of intricacy for climate research. The ICON-Sapphire setup has been expanded to include 600 AMD nodes, resulting in a total of 76,800 cores (Giorgetta et al., 2022; Wedi et al., 2020). This configuration has successfully achieved simulation speeds of 126 SDPD (simulated days per day) at a resolution of 5 km. This demonstrates the considerable capability of this paradigm on high-performance computing platforms. Nevertheless, ICON-Sapphire has showcased commendably swift SDPD emulation speeds on both CPU and GPU platforms, albeit its present execution does have certain constraints. Specifically, it disregards significant parameterization schemes, such as the deep convection scheme, which can greatly influence simulation outcomes, even when using higher resolutions and non-hydrostatic conditions.






Most of the existing work still mainly targets at some parts of the models or years of journey to making significant code changes for porting and optimization, which are still far from the level of complexity of global atmospheric models. Besides, the scientific team continues the effort for developing ultra-high-resolution models based on the new version of the code, even with only a modest capability to accomplish simulation of days to weeks, which becomes a new task as challenging as the beginning point and requires significant code changes. The performance tuning effort came to a stall for the new version. So it's better for the community to take a automated approach that reduces the workload. With some fortune, we succeed, here managing to report results of some concrete progress.

In this study, we describe a mostly automated approach to port the extensively applied Earth system model CESM 2.0 (Danabasoglu et al., 2020), to a new Sunway supercomputer equipped with around 6500,000 SW26010P cores (see section 2.1 for more details). We maintain the framework of CAM-SE while adopting component resolutions and physical models that are highly consistent with scientific computing. This ensures that our research and optimizations can be efficiently transferred to scientific computing applications. Instead, we employ an integrated approach that combines OpenMP and Athread and reconfigure the MPI initialization communication scheme for a hierarchical grid system, employing complementary tools like SWLU to ensure successful code migration. In this way, we effectively enhance the performance of various high-resolution ESMs on the new Sunway supercomputer, achieving atmosphere resolutions up to 5 km and ocean resolutions up to 3 km (Caldwell et al., 2021). Our ported coupled model achieves remarkable scalability, successfully running on 610,800 processors of the Sunway supercomputer, with a simulation performance of 222 SDPD. This performance is competitive when compared to other leading modeling efforts such as COSMO (Fuhrer et al., 2018), HOMMEXX-NH (Bertagna et al., 2020), NICAM (Yashiro et al., 2020), and ICON (Hohenegger et al., 2023), which reported SDPD values ranging from 2.5 to 354.

After the introduction, we describe the significant features of the series of HRESMs on the new Sunway heterogeneous supercomputer in section 2. Section 3 gives details of our new strategy for massively refactoring and optimizing HRESM on the Sunway heterogeneous supercomputer. While stable and sound scientific results of optimized SW-HRESMs are demonstrated in section 4, summary and discussions are given in section 5 finally.

## 2 A series of HRESMs developed on the new Sunway heterogeneous supercomputer

### 2.1 The new Sunway supercomputer system

As the successor of the Sunway TaihuLight supercomputer, the new Sunway Supercomputer provides a peak performance of 1.5 Eflops thanks to a larger number of chips and the improved high-performance many-core processor. The new Sunway supercomputer system is made up of the latest generation of the Sunway supercomputer node called SW26010P, and the total number of nodes is more than 100,000.

The SW26010P node uses the on-chip heterogeneous architecture and is composed of 390 cores, which are divided into 6 core groups (CGs) connecting via a network-on-chip (NoC). Each CG consists of a management-processing element (MPE)





and 64 computing processing elements (CPE) organized in an 8×8 grid. The MPE and CPE both provide support for SIMD instructions, with the MPE supporting 256-bit SIMD instructions and the CPE supporting 512-bit SIMD instructions.

Regarding the memory architecture, the SW26010P is equipped with 96 GB of DDR4 memory, which is evenly distributed to six CGs. In addition to the 32KB L1 instruction cache, the CPE provides 256KB of Local Data Memory (LDM), which is a scratchpad memory that can be directly controlled by the user. The MPE and the CPEs within a CG share the same main memory, and CPEs can directly discretely access main memory through gld/gst or direct memory access (DMA) to transfer the data between main memory and LDM. In each CG, four neighbouring CPEs are interconnected via a single router, which mediates the exchange of contiguous data chunks. In theory, DMA offers a bandwidth of 307 GB/s, while RMA reaches 460 GB/s. Although the storage of LDM increased, it is still too small to put all the necessary data in at most times, so programmers have to do data blocking and copy data in or out of LDM frequently. To best utilize the LDM is the crucial key to achieving better performance for various applications.

To utilize Sunway Accelerate Computing Architecture (SACA) for the series of HRESMs on the new Sunway heterogeneous supercomputer, we propose a three-level parallelization scheme, including process level parallelism implemented on MPE, parallelism implemented on CPEs, and SIMD acceleration closely combined with the underlying hardware.

## 145 2.2 The CESM2 framework with updated oceanic gridding system sustainable for high-precision ocean models

## 2.2.1 The CESM2 framework

The CESM2 displays progressive and distinct features in comparison to the previous version of CESM1.3, as the updated version includes several enhancements in atmospheric and oceanic physics (Danabasoglu et al., 2020). CESM1.3 was previously utilized in the creation of the coupled 25v10 ESM (Zhang et al., 2020; Chang et al., 2020). The updated version incorporates enhancements across multiple components, encompassing the atmosphere, land, ocean, and sea ice.

As a crucial component of CESM2, the Community Atmosphere Model (CAM) has recently undergone an upgrade to version 6, and significant progress has been achieved thanks to the enhanced microphysics scheme and the novel parameterization implemented for subgrid orographic drag. The current CAM in CESM2 consists of 70 vertical layers, reaching a maximum height of  $4.5 \times 10^{-6}$  hPa, which is approximately 130 km. This enables enhanced resolution of atmospheric systems at various scales, such as tropical cyclones and oceanic mesoscale eddies. The CLUBB (Cloud Layers Unified By Binormals) scheme (Golaz et al., 2002; Larson, 2017), which aims to tackle turbulence issues in a comprehensive manner, has also been incorporated into the latest version. The ocean component of POP2 (the 2<sup>nd</sup> version of Parallel Ocean Program) proposes significant advancements in both physical and numerical aspects as novel parameterizations devised to account for mixing effects in estuaries with the Langmuir mixing model and the wave model component (Sun et al., 2019; Li et al., 2016). The sea ice component incorporates CICE Version 5.1.2 (CICE5) (Hunke et al., 2015) and doubles the vertical resolution by







increasing the number of layers from four in CESM1.3 to eight in CESM2. This issue significantly enhances the model's capacity to resolve salinity and temperature profiles.

Progress is also made in the land component. Significant improvements have been achieved in the fifth version of the Community Land Model (CLM5) (Lawrence et al., 2019), with land components updated for better utilization in several areas. Firstly, CLM5 has enhanced its depiction of hydrological, ecological, and climate effects by intensifying the incorporation of human land use activities and incorporating land ice models (Hurrell et al., 2013; Lipscomb et al., 2013). Thanks to some scientific breakthroughs, CLM5 has made extensive progress in describing plant nitrogen dynamics compared to previous versions. Firstly, CLM5 introduces a new plant model that enhances the characterization accuracy of leaf and root physiology (Danabasoglu et al., 2020; Lawrence et al., 2019; Fisher et al., 2019), primarily through the incorporation of plant nitrogen cycle mechanisms and nitrogen uptake traits, showing an enhanced capability in reproducing Gross primary production, leaf area index and biomass (Ghimire et al., 2016). Besides, CLM5 abandons the standard parameterization technique of plant photosynthetic capability and instead embraces the Leaf Utilization of Nitrogen for Assimilation (LUNA) mechanism model (Ali et al., 2016). This innovation allows the model to calculate the response of photosynthetic capacity to environmental variables more accurately. Thus, it might more reasonably depict the actual state of plant photosynthesis. In addition, CLM5 overcomes the inadequacies of terrestrial biosphere models in predicting the carbon costs of plant nutrient collection. The newly incorporated Fixation and Uptake of Nitrogen (FUN) model incorporates carbon cost as a root type function during nitrogen acquisition, thus boosting the model's prediction accuracy for nitrogen absorption and carbon sink (Brzostek et al., 2014; Shi et al., 2016). In summary, compared to observation data, CLM5 is more sensitive to CO2 and other greenhouse gases in terms of carbon absorption, whereas its response to nitrogen fertilizer is rather moderate. This innovation enables CLM5 to replicate the global carbon cycle more closely to the real observation data, thus boosting the application value of the model in the research of global climate change (Lawrence et al., 2019; Fisher et al., 2019; Wieder et al., 2019).

Moreover, a tool called Common Infrastructure for Modeling the Earth (CIME) is introduced into the coupler to send status messages and fluxes between different components in the model to support land-ice coupling and some data overwriting functions (Danabasoglu et al., 2020).

#### 2.2.2 A new TS grid system for kilometer-level high-resolution ocean and sea-ice models

In this study, we utilize a new tripolar ocean/sea ice grid system known as the Schwarz-Christoffel conformal mapping (i.e., TS grid system) (Xu et al., 2021). This grid system allows us to create ocean and sea ice model components with a resolution at the kilometer level. The grid system demonstrates exceptional performance in the three-pole grid arrangement. The new TS grid system offers a notable benefit over older three-pole grid systems like TX0.1V2 (McClean et al., 2011; Murray, 1996) when it comes to addressing the transition zone issue that arises when the latitude and longitude lines converge on the three-pole of the northern patch (NP). In a conventional grid arrangement, the convergence of "latitude and longitude" lines at three poles typically results in an uneven distribution of Height of T-cell East (HTE) in the transition zone. The new TS grid system enhances the scalability and flexibility of the grid system by employing advanced NP stereoscopic projection, resulting in a








smoother distribution of HTE in the transition zone. The TS grid system has been effectively incorporated into the CICE and POP components of the new CESM framework (Xu et al., 2021), establishing a strong basis for our study. We have conducted tests on four tripolar grids with varying resolutions, namely TS015, TS010, TS005, and TS003. These grids correspond to nominal resolutions of around 15, 10, 5, and 3km, respectively. Furthermore, we have incorporated intricate characteristics of the seabed topography to enhance the accuracy of simulating the aquatic environment. The utilization of high-resolution grids and comprehensive seabed topographic data enables us to carry out broad investigations using climate models and thoroughly examine ocean phenomena at a sub-mesoscale level.

## 2.3 A series of HRESMs: 25v10, 12v5, 9v5, 5v5, 5v3 models

For creating a coupled climate model with kilometer-scale resolution, we enhance the existing CESM framework by incorporating kilometer-scale grids in its atmospheric and oceanic components. A sequence of resolutions establishes a hierarchical structure of models encompassing the atmosphere, ocean, and their interactions. For the atmospheric component, i.e. the CAM-SE of CESM, we create a series of quasi-uniform grids with resolutions ranging from 0.25° to 0.06° using cubic sphere grids NE120, NE240, and NE480 (see Zhang et al., 2023 for detailed description), forming a hierarchical structure. It is worth to mention that NE480 achieves a resolution of 5 km, which is four times higher than our prior work (Zhang et al., 2020). The generation of grids is accomplished by the utilization of a novel tripolar grid technique that relies on conformal mapping, namely the TS grid approach, for the ocean and sea ice components. The self-embedding grid hierarchy covers a broad spectrum of ocean resolutions that may resolve mesoscale features, ranging from 0.15° (TS015) to 0.03° (TS003). In order to tackle challenges and detect concerns during the development of HR coupled models, we establish a sequence of resolutions in one component (either atmosphere or ocean) while remaining a same resolution of the other. For example, the NE480 CAM-SE is connected to ocean components with several resolutions at 0.15°, 0.1°, 0.05°, and 0.03°, which are referred to as 5v15, 5v10, 5v5, and 5v3, respectively. In contrast, the ocean portion of POP operating at a resolution of 0.05° is integrated with atmospheric components at resolutions of 25 km (NE120), 12 km (NE240), and 9 km (NE360), denoted as 25v5, 12v5, and 9v5, respectively. Such resolution hierarchy allows us to examine cross-resolution studies pertaining to model adjustment, validation, and processes that are reliant on resolution or scale (Zhang et al., 2023; Xu et al., 2021).

We utilize the remapping feature of the Earth System Modeling Framework (ESMF) tool to generate the mapping coefficient matrices for flux exchanges between the atmosphere and ocean. The coupling of atmosphere and ocean grids is achieved by utilizaing the CESM's standard protocol. This protocol differentiates between flux coupling (conservative mapping), atmospheric momentum forcing (high-order mapping), and state variables (bilinear mapping). With a 6 km nominal resolution, CAM-SE reaches a scale at which nonhydrostatic atmosphere effects become perceptible. However, the model's effective resolving scales are usually 4 times of grid distance. This exceeds the applicable range for nonhydrostatic processes. Therefore, we utilize the default dynamic core of CAM-SE for settings of all models in this investigation. When planning for future configurations with a resolution of 1 km, it is important to take into account the non-hydrostatic effects. This can be observed by utilizing upgraded iterations of the dynamic core.

It's worth to mention that the resolution hierarchy in CESM components facilitates the advancement of the fina coupled model with ultra-high resolution. Atmosphere-only and ocean-sea ice-coupled runs are configured and tested separately through forced trials. The completely connected runs are initiated by the spun-up condition of these components.

#### 230 2.4 Features and performance of 25v10, 12v5, 9v5, 5v5, 5v3 models in MPEs

Districting from the autonomous atmospheric model, ocean model, sea ice model, and land surface model, the CESM propels a complex nonlinear system. In CESM, each model component proceeds autonomous evolution and modification, as well as seamless data interchange. So the alterations in the entire system are reliant not just on the relatively autonomous evolution and modification of each model component, but on their interrelationships and the collective behavior they create. To optimize the future global performance of CESM, we first implement a serial arrangement for pattern components. The statistical running time of the pattern is determined by adding up the running time of each pattern component.

| grids\ Models | 25v10     | 12v5      | 9v5       | 5v5       | 5v3        |
|---------------|-----------|-----------|-----------|-----------|------------|
| Atm/Ind       | 777600    | 3110400   | 777600    | 12441600  | 12441600   |
| Ocn/ice       | 3600*2400 | 7200*5040 | 7200*5040 | 7200*5040 | 12000*8400 |

Table 1: The grid system of the series of developed coupled models.

Figure 1: The runtime percentage and SDPD of model components with different resolutions.

The grid networks and runtime percentage and SDPD of the series of generated grid coupled models of grid networks are displayed in Table 1 and Fig. 1. In the CESM, ATM, OCN, ICE, and CPL components accounts for approximately 90% of the total time, so they play a crucial role in the overall performance of the CESM. Nevertheless, the LND model is a standard computation performed in a single column, and its operational duration is significantly shorter compared with others. Therefore,








we focus our efforts on the ATM, OCN, ICE, and CPL component optimizations and ignore the impact of LND. It is also worth mentioning that, as the grid resolution rises, the initializing time grows exponentially.

Theoretically, increasing the resolution of the atmospheric grid from 25 km to 5 km results in the computational efficiency of the ATM model decreasing around 25 times. Increasing the resolution of the ocean grid from 10 km to 3 km, the computational efficiency of the POP model should decrease around 10 times. The simulation speed is inversely proportional to the square of the grid resolution. The features and performance of the series 25V10~5v3 tests show that ATM simulation is basically consistent with theory. As the atmospheric resolution increases from 12 km to 9 km, the ATM simulation rate decreases by approximately 1.7 times, while maintaining a similar number of processes.

Ideally, if the resolution of OCN and ICE remains constant, the simulation rate of the approximation process should be essentially identical. However, according to the tests, the simulation speed between 12v5 and 9v5 tests is almost two times the difference using 7200 and 7800 processes. The main reason for this situation is that the computational workload of grid blocks and grid blocks in process may still change significantly even if the processes are similar when we choose an automatic domain decomposition scheme. Therefore, the load balancing of domain decomposition is particularly important in POP.

CESM has significant flexibility with respect to the layout of components across different hardware processors. In general, we first drive runs on the union of all processors and control the sequencing with various high-resolution models, then determine the individual performance of each module and modify our PE layout to use a common load balanced configuration for CESM2.2 on the New Sunway supercomputer system.

# 3 A new strategy for highly-efficient optimization of massive scientific programs on heterogeneous many-core super-computers

## 3.1 An overview of computational features of 25v10, 12v5, 9v5, 5v5, 5v3 HRESMs

In CESM, each component is associated with a unique MPI communicator, enabling a stepwise progression toward the ultimate ultra-high-resolution coupled model. After dissecting the features and performance of the series of HRESMs and components, we conduct a detailed analysis of these components, including the atmospheric model, oceanic model, and sea ice model.

The CAM model includes three parts: atmospheric dynamics, atmospheric physics, and communication transfer variables between atmospheric dynamics and atmospheric physics. The atmospheric dynamics is responsible for solving the 3D atmospheric motion equation, and the atmospheric physics is responsible for integrating the physical rainfall, cloud, and radiation in the vertical dimension. The atmospheric dynamics primarily handle the temporal and horizontal iteration of physical values and proceed stencil calculation using the nearby elements. Stencil calculation is the most time-consuming aspect and is heavily memory limited. Different from the atmospheric dynamics, which can parallelly divide into multiple vertical layers, the vertical dimension usually has a strong correlation in the simulation of atmospheric physics. Therefore, we assign balance columns that are further divided into a 2D grid at each level, and multiple layers in the MPE Level, and proceed thread acceleration in a column to simulate the atmospheric dynamics. We also aggregate columns with similar space into the





same process with fewer boundaries and less communication. For the atmospheric physics, we choose the highly parallel columns for load balancing optimization and thread acceleration as the computational efficiency of the atmospheric physics is only affected by the load imbalance between different columns.

The POP adopts a 2D domain decomposition in the horizontal to mesh the grid to the processors. The entire domain is partitioned into equally sized blocks. Typically, the size of a block is represented by "block\_size\_x\*block\_size\_y\*km", where "km" is the number of vertical levels. In POP, integration is divided into barotropic and baroclinic parts. Barotropics use the conjugate gradient method to solve large-scale sparse linear equations, mainly for two-dimensional calculation and neighbor communication. The baroclinic uses the explicit frog leaping integral method to solve the three-dimensional fluid equation, accompanied by a small amount of neighbor communication. So the coupling of baroclinic calculation is closely related to the grid quantity in the process, while the barotropic solution process is affected by the global communication. During thread acceleration, we divide the core computing in pop into two forms: (1) vertical dimension independent calculation and (2) vertical dimension cumulative calculation. We develop distinct algorithms for vertical and horizontal mixers while doing these two sorts of calculations. Additionally, in order to optimize the utilization of multi-core acceleration, we strive to guarantee that the value of POP\_BLCKY is a little less than the core group numbs (64).

The dynamic changing characteristics are the main factor affecting the calculation and load balance of CICE. Similar to the POP, the block is also the smallest unit for domain decomposing. But the connection between surrounding elements in the sea ice model occurs more frequently due to the formation and breakage of ice during the simulation. Meanwhile, there is extensive point-to-point connection in border data communication. In CICE, only these blocks with sea ice distribution are taken into computation, and the communication at the halo boundaries is more disturbing. Regional decomposition is a coordination between computation and communication. For thread acceleration, we first choose better block balanced communication and proceed with thread acceleration according to the number of blocks.

LND presents a typical calculation form called single column form without communication between vertical column grids, and its spatial heterogeneity is enabled by a layered subgrid. Although the single column form highlights the strong nonlinear characteristics, the computational time required for LND is not as significant as that of the ATM, POP, and CICE, as the number of variables in LND is much fewer. So we focus our efforts on optimization to achieve better parallelism and good balance on ATM, POP, and CICE.

## 3.2 Non-intrusive refactoring and fine-grained redesign of HRESMs

This work adopts a non-intrusive porting strategy combined with fine-grained redesign to strike a better balance between code consistency and computational performance. This approach departs from our earlier approach (Fu et al., 2017), which involves time-consuming manual rewriting and tuning of a large volume of code. Since the SW26010P's Sunway Compiler is GCC-based, it simplifies interfacing with libgom, the GNU OpenMP (GOMP) runtime library. O2ATH acts as a bridge between Sunway's Athread Library and the GOMP library, using a custom interfacing mechanism to enable seamless translation of calls. Therefore, our optimization process consists of two steps:



(1) Swift and thorough coverage of OpenMP thread acceleration using O2ATH. The broad applicability of this method is crucial for porting Earth system models. These models are highly constrained by Amdahl's Law because they contain numerous kernels representing different climate processes. Therefore, optimizing only a subset of components typically brings limited performance gains (Satoh et al., 2017). OpenMP (Freitas et al., 2020) offers a precise and detailed representation of parallel algorithms. By using accelerated "pragma" instructions in the source code, the compiler can automatically parallelize the application and incorporate synchronous mutual exclusion and communication as needed to achieve thread acceleration. The computational core automatically enhances thread performance by including OpenMP acceleration instructions through the compiler. There will be a decrease in the number of modifications and restructurings in the code, leading to a substantial reduction in the workload associated with coding. An illustrative instance for MPE source and CPE source coupling using the O2ATH is depicted in Fig. 2 (a). The !\$omp target and !\$omp end target directives mark the bounds of the parallel regions according to the calculation characteristics of each ESM module. For any function called within such a region (e.g., exam), its definition requires the inclusion of the !\$omp declare target directive.

(a)

(b)



Figure 2: Non-intrusive refactoring and fine-grained redesign strategies. (a) The workflow for MPE source and CPE source coupling when using the O2ATH; (b) Subroutine or illustrative instance using Athread and OpenMP.

(2) The Athread fine-grained optimization. Through an OpenMP based source-to-source refactoring tool, we get automated and efficient porting for all major CESM modules to the SW26010P architecture. Then we carry out a more assertive and detailed redesign approach for the most computationally intensive kernels using fine-grained optimization to break down constraints of OpenMP for critical kernels. We take advantage of Sunway's architecture rather than just making use of the parallelism itself. Athread (Fu et al., 2016) is a specialized thread library for Sunway processors that facilitates efficient control and scheduling of threads within CGs, and enables flexible customization acceleration and vectorized arithmetic operations. It is advantageous to modify the placement of Athread\_spawn and Athread\_join in order to effectively coordinate and parallelize tasks across the MPE and CPEs. The fine-grained Athread approach requires both the invocation from the slave core and the transfer of data from the master to the slave to be done manually. The illustrative subroutine or illustrative instance using Athread and OpenMP is depicted in Fig. 2 (b).

## 3.3 O2ATH: Proxy Toolkit Enabling OpenMP Offload on SW26010P

Our O2ATH toolkit comprises a compiler plugin together with a runtime library. As Fig. 2(a) shows, the plugin identifies the target processing element (CPE or MPE) for each function and removes redundant functions from the compiler IR. Utilizing GOMP, the O2ATH plugin constructs the initial OpenMP-parallelized IR, which is then separately processed and optimized for the MPE and CPE objects, with their respective object files ultimately being linked to form the executable file. O2ATH algorithmically selects the appropriate compilation target (CPE or MPE) for functions and eliminates redundant functions in


the compiler IR. As illustrated in Fig. 3, a set of fundamental OpenMP constructs in the user code box are supported by the runtime library.

Figure 3: The operational workflow for executing tasks when utilizing the O2ATH runtime library.

The primary impediment to implementing OpenMP capabilities using CPE threads is the synchronization construction.

These constructs are typically employed to ensure data consistency. Due to the incomplete maintenance of cache coherency

by CPEs, we provide explicit cache flush operations to facilitate these structures.

The SW26010P heterogeneous many-core processor with its distinct memory model poses another challenge compared to conventional OpenMP offloading platforms. The LDM of the SW26010P is significantly reduced in size, by several orders of magnitude, compared to Graphics Double Data Rate (GDDR) or High-Bandwidth Memory (HBM). To address this issue, our current strategy involves leveraging LDM as the CPE stack when its capacity is sufficient. In cases where LDM is insufficient, we allocate the stack within the private memory space of CPE. We develop a wrapper script that compiles a single file twice to produce CPE machine code, thereby overcoming the limitation of hybrid-LTO being unavailable in swgcc.

Optimizations performed with O2ATH do not interfere with low-level tuning. Its usage is similar to SWACC, but the compilation speed is increased by roughly tenfold, significantly reducing the time required to recompile the CESM code. As seen in Fig. 4, the compiling script xfort.py comprises many essential phases to guarantee the compilation process is efficient. First, xfort.py analyzes the compilation commands and organizes the arguments into three distinct groups: args\_host, args\_slave, and args\_common. Additionally, if the option to leave the source file is selected, xfort.py scans for O2ATH entries inside the source file. These entries define CPE functions for making function calls across files, enabling smooth integration.

Furthermore, xfort.py builds the MPE object while enabling the compiler plugin. The plugin analyzes the call graph to execute supplementary tasks throughtout this process. When xfort.py detects references to SLAVE routines, it triggers the compiler to create CPE code. Ultimately, the MPE and CPE objects are connected inside the local environment to provide the ultimate compilation result. If xfort.py does not find any SLAVE references, it considers the MPE object as the destination file. The organized approach guarantees a methodical and effective compilation process, making full use of the capabilities of the O2ATH plugin.

Figure 4: The workflow of xfort when the plugin is used to compile source files.

With the OpenMP directive and O2ATH, tasks can be offloaded to the CPEs. With appropriate offloading approaches, extensive and efficient parallelization can be realized across distinct component models, leading to significant performance improvements. Here are some guidelines for adopting OpenMP:

- Add as few private variables as possible because of limited memory.
- Analyze variable attributes and look for variables with write conflicts.
- In the loop area, local variables that need to be assigned and will be used in the subsequent loop process generally need to be local variables with limited dimensions in the private loop area. Generally, priority is given to whether they need to be set to private.


• Use vnest\_element\_prefix\_sum() in terms of the "cumulative sum" class loop, and add call flush\_slave\_cache(), call vnest\_syn(). Plus, it ensures that parallelism and serial have the same computation order to solve the "sum-sum" parallel precision problem.

#### 3.4 Athread-based fine-grained redesign

Utilizing OpenMP thread acceleration is essential in the effort to improve the performance of the Community Earth System Model (CESM). However, some crucial operations are unable to fully take advantage of this strategy because of constraints in processors with heterogeneous architecture. In order to tackle this issue, we have implemented a more sophisticated MPI+Athread programming paradigm for a few major kernels, with the goal of achieving exceptional acceleration performance.

The brief refactoring and redesigning of our Athread-based fine-grained effort are:

- (1) Code refactoring, including Fortran to C, more reasonable data layout, data merging, etc.
- (2) RMA Communication-based Parallelization Scheme for data reuse between the LDMs.
- 385 (3) Fine-grained load balancing at the parallel level of the slave core is achieved by cycle modification, segmentation, and merging.
  - (4) Manual vectorizations. The translation from Fortran to C establishes the basis for implementing vectorization in data structures.
    - (5) Double buffering to overlap the computation and the communication.
- We select several identical kernels to compare the speed up of Athread and OpenMP optimization, as shown in Fig. 5. The Athread manual implementation can achieve better acceleration for kernels such as euler\_step, vmix\_kpp, vertical\_mix, and baroclinic.

Figure 5: Performances of CAM6 and POP2 with O2ATH and manual optimization.





## 395 3.5 Developed tools and initialization boosting

In order to lower the manual code workload and to fully exploit the advantages offered by software tools, we have concentrated much of our effort on upgrading current tools and creating new ones.

A collection of tools is employed to facilitate performance evaluation and loop tuning during program porting, including SWMU for memory diagnostics and content inspection; SWLU, a sampling-based performance profiling tool; libvnest, a library designed to support nested vertical-level parallelism for dynamic cores; as well as other tools for performance monitoring, debugging, and diagnostics, such as SWCallgraph, GPTL, and BinaryCheck.

Memory requirements during initialization are substantially reduced by restructuring the MPI communication into a hierarchical Allto Allw approach, which also successfully prevents memory explosion at scale. Besides, a reduction in the complexity of mapping physical node IDs to MPI ranks from quadratic to logarithmic order is achieved by employing a quick sort algorithm based on fixed-length strings. Furthermore, we change the communication of the CICE from block-wise communication to a packed MPI gather communication. I/O balancing and redundancy reduction are also implemented to further shorten initialization time.

## 3.6 Numerical precision verification after CPE parallelization

It is crucial to verify models during the optimization and development of Earth System Models (ESM) in order to build and maintain their credibility. The purpose of this method is to verify that the implementation of a model is precise and consistent with its original description and assumptions. A conventional approach involves comparing identical simulation data on both the new machine and the dependable machine, with the simulation often spanning several hundred years, usually about 400 years. Subsequently, several approaches that assess consistency using ensembles are used to compare the new simulations with the control ensemble generated by the reliable machine.

(1) Master and slave core binary consistency tool

The main concept of this approach is to provide binary consistency between the control cores of standard processors and diverse supercomputers, as well as to attain binary consistency across instruction set architectures and compilers. The implementation procedure comprises three sequential steps:

- Identify the compilation choices. To ensure binary consistency, it is necessary to identify a compilation option for each combination of instruction set architectures and compilers. The compilation settings should prioritize program performance to the greatest extent feasible without compromising correctness.
- Remove the impact of external libraries. Utilize the compilation parameters obtained from the previous stage to recompile external libraries that have an impact on correctness, such as the math library and the Fortran standard library. Verify that these libraries adhere to binary consistency criteria.

- Nullifies the impact of the compiler's inherent optimizations. During the compilation process, the compiler typically
  maximizes optimization by using built-in instructions and deriving constants. The handling of these inherent optimizations varies across various compilers and can impact the binary consistency of the outcomes.
  - (2) An approach for consistency testing based on deep learning

We have developed ESM-DCT, an advanced consistency testing method for ESMs that use deep learning techniques. This tool includes three parts:

- Creating an unsupervised model using the BGRU-AE model. The purpose of this model is to assess non-linear patterns seen in atmospheric and oceanic simulation data.
- Developing an Ensemble Approach: utilizing statistical distributions obtained from short-term simulations of atmospheric and oceanic components as datasets.
- Deploying the software tool: adapting to uncertainty arising from different hardware configurations and evaluating uniformity across several HPC systems.

## 4 Performance results

## 4.1 Performance improvement of kernels on CPEs

Parallelization over the CPE array is the initial step in utilizing the heterogeneous many-core processor. O2ATH and Athread toolkits enable most kernels to be transferred onto the CPE array from the MPE, avoiding additional programming efforts. Subsequently, an intensified and detailed redesign utilizing Athread is carried out to further squeeze out the optimization. The performance is evaluated based on the mean runtime of three repeated executions, considering both timing and flops. Improvements of the key kernels in the atmpsphere and ocean components are illustrated in Fig. 6(a) and (b).

The majority of kernels within the dycore of CAM involve boundary exchange between components. Kernels such as EU-LER, RHS, and HYPERVIS are affected by Amdahl's Law and do not exhibit significant improvements in speed. Key kernels, including LAGRANGE, FVM, and OMEGA, achieve notable acceleration through CPE parallelization. The performance gains are 6 to 9 times, approaching the theoretical memory bandwidth limit. In the dycore, the vertical-layer dependence of the OMEGA kernel can be resolved using the prefix-sum tool provided by libvnest. This approach eliminates the performance penalty caused by data dependency.

An important aspect to note is that our methodology enables the transfer of the complete physics system onto heterogeneous CPE arrays with few changes to the existing code. The physics of the system before and after connection, referred to as BCPHYS and ACPHYS, respectively, see significant speed improvements of up to 6.8 times and 10 times. One potential explanation for the comparatively smaller speed up of BCPHYS might be attributed to the intricate code and subsequent suboptimal cache performance and inaccurate branch predictions.


Figure 6: Performance improvement. (a) Performance improvement for the components in ATM; (b) Performance improvement for the components in OCN; (c) Performance improvement for the components in SW-HRESMS.

In POP2, the speed of both vertical and horizontal parallelism inside blocks is significantly improved by dividing horizontal blocks into a "thin and tall" form.

The performance gains of the full component models are presented in Figure 6(c), highlighting both the dynamic and physical parts of CAM. CAM's physics part achieves the greatest acceleration due to the absence of communication, while CICE is greatly limited by communication overhead. In ultra-high-resolution simulations, CICE performs slower than CAM due to its strong coupling with CAM, and its computational cost increases further as the atmospheric timestep size is reduced. On CPEs, CICE and POP attain speedups of  $2.43 \times$  and  $1.86 \times$ , due to their differing responses to communication and computational load.

After the optimizations, the series of SW-HRESMS achieves an improvement in computing efficiency by five times on average. Notably, the highest resolution 5v3 model now performs at nearly one simulation year per day, compared to 1.7 simulation days per day with the previous version that utilized Management Processing Elements only.




## 4.2 Important milestones

We illustrate important milestones after scaling the model to 300K CGs in Fig. 7. At very beginning, we underestimated the computation cost in high-resolution CICE, causing CICE to have a large occupation in model run time. With the optimizations in CAM, the simulation performance increases steadily, causing CICE and POP to become the bottlenecks of overall performance. CICE is optimized by making use of inter-block optimization with CPEs and achieves a speed up of 2.43 times. The implementation of a multi-level MPI collective communication library resolves the constraint on connection counts, enabling model expansion to 460,800 CGs. However, this scaling reveals a performance bottleneck in the CPL, which is diagnosed by SWLU as an inefficient MPI All-to-All routine. To address this, the MCT is redesigned to employ a send-recv pattern for data rearrangement. Subsequent optimization of POP on the CPEs delivers a final throughput of 222 SDPD. We make efforts to optimize the initialization procedure of CESM. First of all, we need to adjust PIO settings to make the initialization procedure runnable with 300K core groups. But there is a significant performance loss due to such adjustments and the increment of process count. After that, we replace PIO1 with a hierarchical MPI library powered PIO2, and the initialization procedure only costs 0.92 hours. Also, there is some legacy code that is not well optimized for large-scale runs, such as broadcasting variables in-turn or nested process iterations. These problems are identified with SWLU and optimized one-by-one, making the initialization take only 0.58 hours.

Figure 7: Important milestones in model run.

## 4.3 Scaling results of CAM, POP, and the entire SW-HRESMS

Figure 8 (a) and (b) demonstrate the weak scaling outcomes of CAM and POP. These tests are conducted under the principle of a consistent timestep and an unchanged grid-point-to-processor ratio between the low- and high-resolution configurations. As communication transitions from within a single cabinet (ne30) to across different cabinets (ne120), CAM's parallel efficiency drops to 77%. A further decline to approximately 72% occurs when scaling beyond half of the full capacity. POP



experiences comparable decreases in parallel efficiency during communication mode switches and with rised parallel size. However, its performance often surpasses CAM's because POP is more efficient in utilizing computing nodes.

Fig. 8 (c-e) presents the results of strong scaling tests. The smallest feasible scale for NE480 is 28,800 processes, allowing for the simulation of just 41.28 model days each wall day. The scalability of the NE480 simulation, measured as the ratio of performance improvement to the increase in the number of processes, is around 47% when scaling from 14,400 processes to 460,800 processes. The scaling efficiency of POP is measured at 9.59 SDPD, and it demonstrates a modest scaling efficiency of 70% on the TS003 mesh. This is because the majority of our parallel techniques are not dependent on a singular dimension. Therefore, the CPEs are capable of enhancing calculation speed in the strong scaling situation. We successfully executed the fully integrated simulation using CAM, POP, CICE, and CLM, all based on the CPL7 framework. The peak performance achieved by the connected model is about 50 SDPD, utilizing one-fourth of the computational power of the new Sunway supercomputer. This performance level surpasses the result obtained while using the entire machine, which is 30 SDPD. The present scaling efficiency of CPL7 and CICE is unsatisfactory. One possible explanation might be the incorrect setup of cases. By fine-tuning the coupler and CICE at the machine's full size, it is possible to achieve significant improvements in the expected outcomes of full-coupled performance, reaching around 222 SDPD.

(e)





Figure 8: Scaling results of CAM, POP, and the entire model. (a) Weak scalability for ATM; (b) Weak scalability for OCN; (c) Strong scalability for ATM; (d) Strong scalability for POP; (e) Strong scalability for the SW-HRESMS.

#### 4.4 Preliminary simulation results of SW-HRESMs

The sequence of high-resolution models optimized above are initialized from the coupled states remapped from the long time pre-industrial control run of the previous 0.25° atmosphere and 0.1° ocean resolution coupled model (the old version of 25v10) (Chang et al., 2020). These model are integrated for about 1 year, and the results after initial spin-up processes are used for cross resolution inter-comparisons and the study of various processes. Key characteristics of the coupled climate system are all well characterized with a hierarchy of model resolutions, including tropical cyclones (TC), mesoscale/submesoscale modulated air-sea interactions, and topography/bathymetry effects on geophysical fluid. For example, slightly stronger TCs are simulated with 5 km-resolution resolution than 25 km resolution, but the modeled structure of the TCs, including the eyewall and wind/precipitation bands, is much finer in the former (Zhang et al., 2023). Also, storm track is much stronger due to finer sea-surface temperature gradients and ensuing air-sea interactions enabled at submesoscale-permitting resolution of TS003. Figure 9 shows the snapshot of the ocean surface's vorticity and sea ice deformation during winter in the Southern Oceans for the coupled run with NE480-atmosphere coupled with TS003-ocean (i.e. the 5v3 model). Highly ageostrophic, submesoscale-rich oceanic flow is prevalent in key air-sea interaction regions of subtropical gyres and ACC (Antarctic circumpolar current). Linear kinematic features, which are unique to sea ice dynamics, arise due to multiscale, fractal plastic deformations, which have profound effects on moisture and buoyancy fluxes across the air-sea interface. We show the surface pressure P<sub>s</sub> fields produced by 12v5, 9v5 and SST produced by 5v5, 5v3 in Fig. 10. As a unique characteristic of our approach, the resolution hierarchy in both the atmospheric and the oceanic components enables us the cross-resolution/scale study of the modeled processes. For example, with the ocean resolution increases to 0.03° from lower bound, the atmospherically forced ocean simulations witness over 90% increase in the total ocean kinematic energy. Although at 0.05° resolution (4 km global average), the ocean is capable to resolve certain submesoscales, finer structures such as ocean fronts are much better characterized at the 0.03° resolution. Coupling with the atmospheric model with NE480 introduces much higher frequency variability for the ocean's forcing, resulting in further increases of kinetic energy.

Figure 9: Ocean surface relative vorticity ( $\zeta$ ) and sea ice concentration/deformation of wintertime southern (left) and northern (right) hemisphere for the ultra-high resolution coupled run (5v3 - NE480 coupled with TS003).







Figure 10: The *ab*) surface pressure (P<sub>s</sub>) and *cd*) sea surface temperature (SST) of NE240- and NE360-atmosphere coupled with TS005-ocean (12v5 - panel *a* and 9v5 - panel *b*) as well as NE480-atmosphere coupled with TS005-and TS003-ocean (5v5 - panel c and 5v3 - panel d) after integration for 1 year as they are initialized from 00UTC 1 January 646-year of 25v10 CTL simulation (Chang et al., 2020).

Follow-up numerical integration and further analyses of the model outputs are undergoing. In particular, we plan to further studies on air-sea coupling processes, as well as the reduction of the long-standing biases of climate models in the topics using longer, multi-year's simulation results.

## 5 Summary and discussions

As the scale of operation gradually increases from the hundred-kilometer range to the kilometer level, there is an urgent need for highly efficient optimization of kilometer-level resolution ESMs on heterogeneous many-core systems. In this paper, we introduce an OpenMP-based tool tailored specifically for the new Sunway supercomputer, which facilitates automated code optimization. This approach supports a non-intrusive and broadly effective migration of the CESM 2.0. Our methodology is characterized by minimal manual code alterations, coupled with meticulous redesign, aimed at enhancing both performance and maintaining the consistency of the model code. Leveraging a hierarchical grid system and an OpenMP-based offloading toolkit, our automated porting and parallelization efforts encompass over 80% of the codebase. Our results demonstrate a substantial increase in simulation speed, from 1.1 to 222 SDPD, enabling multi-year or even multi-decadal scientific experiments with high-resolution coupled simulations. This work signifies a pivotal advancement in Earth system modeling, providing a robust platform for HR climate simulations.

Looking ahead, we anticipate further optimizing the CPL7 and CICE components at large-scale runs, potentially enabling simulations of approximately one year per day. With continuous enhancements in resolution, kilometer-scale coupled models will emerge as the cornerstone for numerous scientific endeavours and critical applications related to climate change. They hold the promise of delivering a more nuanced portrayal of multi-scale atmospheric, oceanic, and coupled processes, including finer simulations of the energy cycle and air-sea interactions at frontal scales. The knowledge and data generated from these endeavours will constitute an invaluable resource for climate model development, future projections, and climate mitigation strategies.

The SW-HRESMs possess the capability to capture sub-mesoscale tropical cyclones, ocean mesoscale vortices, and the interactions between vortexes mean flows with a heightened degree of resolution. This achievement paves the way for the further advancement of model development, as well as enhancing the simulation capabilities pertaining to major extreme weather and climate phenomena in both the atmosphere and the ocean. In our 5v3 model, the current horizontal resolutions of the atmospheric and marine components reach 5 km and 3 km, respectively. While this resolution is sufficient to reveal the manifestation of cumulonimbus cells in the atmosphere and sub-mesoscale activities in the ocean, the characteristic horizontal scales of these phenomena, which are approximately 10 km and several km, respectively, are not yet distinctly discernible.




Consequently, seamless weather and climate research necessitates the use of ESMs that are capable of representing multi-scale interactions, particularly those pertaining to energy cascades and inverse cascades associated with finer scales.

In this paper, the primary emphasis of hardware is placed on the Sunway TaihuLight supercomputing system. However, we contend that numerous refactoring and optimizing procedures outlined herein can also prove beneficial to the formulation of code porting and optimization strategies for other heterogeneous many-core systems, notably GPU-based high-performance computing (HPC) systems. As the inaugural modeling endeavor focused on heterogeneous many-core supercomputing systems, this study has not encompassed the broader aspect of general performance portability. In light of the escalating requirement for heterogeneous HPC modeling, the matter of general performance-portability will undoubtedly emerge as a pivotal and intriguing theme in future investigations.

Moreover, we acknowledge the formidable challenge posed by ensuring the accuracy of recoding on a heterogeneous many-core supercomputing platform. To this end, we anticipate embarking on subsequent studies that leverage CESM-ECT for high-resolution simulations in general, with a specific emphasis on its application to the ongoing CPE optimization. Given the swift progression of heterogeneous many-core supercomputing platforms, assessing the efforts directed towards high-resolution code refactoring is of paramount importance to the community at large.

## Code and data availability

The model code for the open-source CESM2 project is publicly available at <a href="https://github.com/ESCOMP/CESM">https://github.com/ESCOMP/CESM</a> (last access: 4 Dec 2024; <a href="https://github.com/ESCOMP/CESM">UCAR/NCAR</a>, 2020). Optimized model code and raw data can be provided by authors upon request.

## **Author contributions**

X.L, Z.L, S.Z, H.F designed and led the research and drafted the manuscript. X.D, Y.L developed the O2ATH and tools, S.X, Y.G, G.Y analyzed the data. All other co-authors were involved in accelerations of the earth system models on heterogeneous many-core supercomputers, and made comments on the manuscript.

Conflict of interest statement. The contact author has declared that none of the authors has any competing interests

## 585 Acknowledgments

The research is supported by National Natural Science Foundation of China (Grant No. T2125006), National Key R&D Program of China (2022YFE0106400), Science and Technology Innovation Projects of Laoshan Laboratory (Nos. LSKJ202300400-03, LSKJ202202200-04), the National Natural Science Foundation of China (42361164616, 42022041, 4228101, 41876001), National Key Research and Development Program (SQ2023YFB3000028, 2022YFB4500501),

Construction of Jiangsu Province Intelligent Computing Application Service Platform, Shandong "Taishan" Scientist Program (ts201712017).

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
