# Peer review of "A highly-efficient automated optimization approach for kilometerlevel resolution Earth system models on heterogeneous many-core supercomputers"

_EGUsphere, 2025_

## Author Comment (AC3)

Dear Reviewer,

Thank you very much for your constructive comments and thoughtful suggestions. We sincerely appreciate the time and effort you devoted to reviewing our manuscript. Your feedback is extremely valuable and provides clear guidance for significantly improving our work.

We have carefully considered all of your points. In the responses below, we address each comment in detail, outlining our understanding and the specific changes we will make to the manuscript to incorporate your suggestions. We are committed to thoroughly revising the manuscript along the lines you have recommended and believe these changes will greatly enhance its clarity, accuracy, and overall impact.

**General Comment**

**Reviewer comment:**

I have a few comments regarding clarity of certain statements/sections, but before that, I think my biggest concern, which makes me think the work may need a major revision, is that there are already efforts in literature that aim to port CESM (including CAM and POP) to run efficiently on the Sunway supercomputer, including previous works by some of the authors in 2016 and 2021. From reading the paper, it seems to me that the authors do not use previous porting as a starting point, but rather do a completely new effort. This seems like a waste, and this choice has to be explained and justified in detail, otherwise this is just presenting a work that is very similar to what was already done. Why not building on previous successes? Would previous work not scale to these resolutions? Or is this in fact the same work, just extended to the CICE and MCT components? Also, how does the speedup of this work compares to the previous porting efforts, if those versions of CESM were to be run at these resolutions? Hopefully, this can be cleared easily by the authors, but the importance of this detail is why I select the "major revision" bullet.

**Response:**

We fully understand the reviewer's concern regarding the relationship between this work and earlier CESM porting efforts, as well as the necessity and differentiation of the present study. To avoid any potential misunderstanding, we provide a clear explanation below addressing why a new porting and optimization effort was required, why previous results could not be directly reused, and how this work fundamentally differs from earlier studies.

First, model version differences prevent direct reuse of previous optimizations. Earlier studies

targeted CESM1.3, whereas the present work is based on CESM2.2. Between these versions, CESM has undergone extensive refactoring and architectural evolution, resulting in substantial changes to the implementation of many core modules. Consequently, the previously refined and optimized code for the old version cannot be directly reused. In this work, only a small fraction of reusable code could be inherited, while most performance-critical components required redevelopment.

Second, the target hardware platform has significantly evolved. Previous efforts were conducted on the first-generation Sunway system, whereas this study targets the new-generation Sunway supercomputer, which introduces important architectural upgrades. Optimization strategies that were tightly coupled to the execution and memory characteristics of the older platform required careful re-evaluation and re-adaptation to the new hardware.

Third, CESM consists of millions of lines of code. Earlier manual Athread-based optimizations focused primarily on a limited number of dominant hotspots. According to Amdahl's law, such partial optimization limits overall performance gains. Achieving further improvements therefore requires expanding optimization coverage to a wider range of computational hotspots.

Based on these considerations, the present work adopts an incremental yet substantially extended optimization strategy. For code sections that remained largely unchanged across versions, we retained the original optimization framework but re-tuned it for the new hardware. For components that underwent significant modifications, we redesigned and re-implemented the optimizations, leveraging the O2ATH tool to cover a much broader portion of the code base.

We have added explanations in the revised manuscript. Please see line 109-117.

**Comment 0**

**Reviewer comment:**

In general, I think the paper could benefit from a grammar/syntax check, especially in the first couple of sections. Most text editors won't flag anything, as the words used do in fact exist (these are not misspellings), but a few more careful reads (or some better grammar/syntax check tool) would prob help identify areas that need fixes. Related to this, NICAM expanded name includes "Iscosajedral", but should be Icosahedral.

**Response:**

We thank the reviewer for pointing out these issues. In the revised manuscript, we have performed thorough grammar and language check throughout the entire text to improve

clarity and readability.

**Comment 1**

**Reviewer comment:**

In the literature review, the authors mention Golaz 2022 as a model that accommodate up to 276,000 GPUs and achieves 0.97 SYPD. However, Golaz 2022 work uses E3SM v2, which is a CPU-only model, and the performance experiments showed in that paper are performed on the Chrysalis supercomputer, located at Argonne National Laboratory. Perhaps the authors were thinking of the Taylor 2023 work (doi: 10.1145/3581784.3627044), which achieved 1.26 SYDP on Frontier, using however only 65,536 AMD GPUs. Also, when comparing the current work's performance with leading modeling efforts, it may be best to not use the Bertagna 2020 work, as that only has the dycore ported to GPU. A better comparison within the E3SM umbrella would be the aforementioned Taylor 2023 paper, or Donahue 2024 (doi: 10.1029/2024MS004314).

**Response:**

Thank you for pointing this out. Due to a technical issue during manuscript preparation, some references in the original literature review were incorrectly associated with the corresponding discussions. In the revised manuscript, we have corrected these citations: the discussion previously attributed to Golaz et al. (2022) now correctly cites Bertagna et al. (2020), and the discussion originally associated with Bertagna et al. (2020) has been corrected to Edwards et al. (DOI: https://doi.org/10.1016/j.jpdc.2014.07.003). We thank the reviewer for this helpful comment.

**Comment 2**

**Reviewer comment:**

In section 2.1, it would be nice to give some more interesting details on the new Sunway supercomputer, or provide in this section a reference to another work with such details. For instance, the clock speed of MPEs vs CPEs, or their nominal power consumption. These details would help putting in perspective the claim done in section 4.1 that "the highest resolution model now performs at nearly one simulation year per day, compared to 1.7 SDPD with the previous version that utilized MPEs only".

**Response:**

We thank the reviewer for pointing out the insufficient description of the new-generation

Sunway supercomputer in Section 2.1. In the revised manuscript, we have added information about the new supercomputer. Please see line 137-139, 143-145, 154-158 in Section 2.1.

**Comment 3**

**Reviewer comment:**

In section 2.4 the authors claim that going from 25km to 5km they would expect a drop in efficiency by a factor 25. However, the factor 25 only accounts for the increase in spatial resolution, while CFL constraints also impose a decrease in the maximum allowed time step. Are the authors using the same dt for all simulations, meaning that they pick a dt that works for the finest resolution, and use that also for coarse resolutions? If so, this should probably be stated, to avoid CFL-related confusion.

**Response:**

We appreciate the reviewer's valuable comment regarding potential confusion arising from CFL constraints. In this work, the comparison focuses on the increase in computational cost per iteration step due to the growth in spatial grid points when the resolution is increased from 25 km to 5 km, without considering adjustments to the time step that might be required for higher resolutions. We have clarified this point in the revised manuscript, please see line 271-279.

**Comment 4**

**Reviewer comment:**

In section 3, I would consider dropping the terminology master/slave, in favor of less "controversial" ones. For instance, they could use "main thread" (or "primary thread") and "worker threads". Other good suggestions can be found browsing the CS community. While "master" is still widely used, the term "slave" is definitely falling out of favor...

**Response:**

We appreciate this suggestion. In the revised manuscript, we have replaced the terms "master/slave" with more neutral and widely accepted terminology, such as "main (or primary) thread" and "worker threads".

**Comment 5**

**Reviewer comment:**

In section 3.4, in particular figure 5, we see the perf boost of individual CAM/POP portions when switching from the O2ATH framework to a pure Athread one. I am not an expert of POP, but the three atm dycore subroutines reported have a different runtime. It may be helpful to add a table (perhaps side by side with the figure) showing the runtime of these portions of the code (for the base case, without either O2ATH nor Athread), to help weigh the different speedup bars.

**Response:**

We thank the reviewer for the suggestion. In the revised manuscript, we have added a table alongside Figure 5 showing the runtime of these code sections under the baseline condition.

**Comment 6**

**Reviewer comment:**

In section 3.6 the authors discuss "binary consistency" across different compilers and/or supercomputers. By this, I assume they mean the bit-wise value of the generated output, and not the binary executable, but perhaps this should be clarified. Either way, I don't think there is much interest across models in retaining a bit-for-bit reproducible solution across different compilers, let alone different machines (but I may be wrong). Usually, this can be achieved for subsequent versions of the code on the SAME machine with the SAME compiler, but that can also be challenging (especially if one uses solvers that do make use of random algorithms, or iterative algorithm involving many global reductions). On the other hand, an ensemble-based Deep Learning approach is very reasonable, and definitely more interesting for the community. Since it seems the authors developed their own framework, I think it would be more interesting to devote the full 3.6 section to this, maybe giving some more detail that can be useful for other centers/models.

**Response:**

We appreciate the reviewer's insightful comments on Section 3.6. First, we clarify that "binary consistency" refers to the consistency of the model's output data, not the executable file itself. Our work mainly involves cross-machine and cross-compiler output data consistency checks to ensure that the optimized model's output remains consistent with the original implementation, guaranteeing correctness even under different hardware and compiler environments. We have added explanations in the revised manuscript. Please see line 477-479.

Detailed descriptions of our self-developed ESM-DCT framework can be found in our previous work (https://doi.org/10.1016/j.isci.2024.111574), which has now been explicitly

cited in the revised manuscript. Therefore, we do not repeat those details here.

**Comment 7**

**Reviewer comment:**

I'm curious as to how the OMEGA kernel acceleration is achieved. Does libvnest implement a divide-and-conquer approach? It would be nice to share a few more details, to allow other projects to learn from this effort.

**Response:**

We thank the reviewer for the interest in the OMEGA kernel acceleration. Currently, the acceleration of the OMEGA kernel primarily relies on the xfort tool to implement worker-thread (CPE) parallelism on the Sunway platform, similar in concept to OpenMP. Specifically, key loops within the kernel are identified, and an appropriate dimension is selected based on data block partitioning principles, with manual task partitioning performed along this dimension. To achieve optimal performance, we should not only consider the parallel dimensions based on the division plan but also the implementation of the loops themselves. In some computations, the most suitable dimension for parallelism is not explicitly present in the loops, and directly parallelizing along the explicit dimension often fails to achieve ideal performance. Therefore, for parts involving implicit operations such as array slicing, we first expand the implicit loops and then apply worker-thread parallelism along that dimension, thereby enhancing the flexibility and overall acceleration efficiency of many-core parallelism. We have added explanations in the revised manuscript. Please see line 512-517.

**Comment 8**

**Reviewer comment:**

In section 4.1 the authors claim that "the highest resolution model now performs at nearly one simulation year per day". In the rest of the 4.x sections the highest number we see (for the full model) is 222 SDPD, which is ~0.6 SYPD. I would not say that 0.6 is "nearly one SYPD"... It is possible (though the paragraph does not suggest that) that they referred to the performance of the ATM component only, which fig 8c shows to achieve 331 SDPD; if so, it should be made clear. If, instead, they were referring to the full model, I would consider rephrasing the claim, as 0.6 is not very close to 1. To be clear, I am not belittling the relevance of the 222 SDPD; I am just saying that I don't see a need to use misleading words.

**Response:**

We thank the reviewer for pointing out the inaccuracy. We confirm that for the full SW-HRESMS model simulation, the actual overall performance achieved is 222 SDPD, approximately 0.6 SYPD, which indeed does not reach the "one year per day" level. We have revised the relevant statement in the manuscript to avoid misleading readers.

**Comment 9**

**Reviewer comment:**

In section 4.2 the plot line shows a 1.83x improve in SDPD in the last optimization step. However, the bar plot of the wall seconds does not seem to decrease by much (if at all). Where does the boost from 131 to 222 SDPD come from, given that the wall seconds bar plots seem unchanged?

**Response:**

We thank the reviewer for pointing out the issues in the figure. Upon checking, the discrepancy between the SDPD improvement and the wall seconds bar plot in Section 4.2 was indeed caused by a plotting error. We have corrected the relevant data and regenerate the figure in the revised manuscript to ensure consistency between SDPD and wall seconds.

**Comment 10.a**

**Reviewer comment:**

In section 4.3:

a) when commenting fig 8, the authors first claim that the smallest feasible scale for NE480 was 28,800 processes (in agreement with fig 8c), but then they say that efficiency is ~47% when scaling from 14,400 to 460,800 processes. So, did 14.400 work? Or is this a typo?

**Response:**

a) We thank the reviewer for pointing out this issue. This was indeed a typo. "14,400" should be corrected to "28,800". We have made correction in the revised manuscript to ensure consistency with Figure 8c.

**Comment 10.b**

**Reviewer comment:**

b) the authors say that "a further decline to ~72% occurs when scaling beyond half of the full

capacity". However, looking at fig 8a, I see an efficiency of 69% at 115,200 CPEs, that is, a lower efficiency, and happening earlier than scaling beyond half capacity. Either this discrepancy comes from data older/newer than the data in the plot, or it refers to different data. Either way, this needs to be clarified/fixed.

**Response:**

b) We thank the reviewer for noting this potential confusion. The "72% parallel efficiency" mentioned in the text comes from experimental results at half-machine scale (around 300,000 processes). This data point lies between the 172,800 processes (75% efficiency) and 345,600 processes (66% efficiency) points on the ATM strong scaling efficiency plot (Figure 8c) but was not plotted. In the revised manuscript, we have corrected the statement to match the plotted results, and the efficiency is now reported as approximately 66%, consistent with Figure 8.

**Comment 10.c**

**Reviewer comment:**

c) the authors say "the efficiency of POP is measured at 9.59 SDPD". I don't understand what it means to measure efficiency at 9.59 SDPD. Also, none of the plots below show the 9.59 SDPD number, so it's hard to link this number to the rest of the section and the plots. This sentence needs to be clarified.

**Response:**

c) We thank the reviewer for pointing out this issue. The "9.59 SDPD" mentioned in the text was a typo. The correct value should be 249 SDPD. We have corrected this in the revised manuscript to ensure consistency between the text and the figures.

**Comment 10.d**

**Reviewer comment:**

d) the authors say that the peek perf is 50 SDPD, achieved with one fourth of the machine, and that using the full machine it drops to 30 SDPD. They speculate that this may be due to "incorrect setup of cases", but it is not clear what they mean by that. Also, these numbers are nowhere in the plots. Where this initial experiments before another round of optimizations? This needs some clarification.

**Response:**

d) We thank the reviewer for the query. The initial peak performance mentioned (50 SDPD with one-fourth of the machine, dropping to 30 SDPD with the full machine) comes from records of the initial experimental testing phase. These data points are not displayed in the figures. In these preliminary experiments, the performance at full machine scale was lower than at one-fourth scale, contrary to theoretical expectations. Analysis revealed that the primary reasons were related to process layout and communication strategies, which led to efficiency decline at large scale. Subsequently, we optimized the process layout and communication, specifically by releasing all processes to CICE and CPL after ATM computation, which significantly improved the overall performance at full machine scale, ultimately reaching about 222 SDPD. We have clarified this description in the revised manuscript. Please see line 568-574.

**Comment 10.e**

**Reviewer comment:**

e) related to the above, the authors then claim that "by fine-tuning CPL and CICE at the machine full size it is possible ot achieve significant improvement". Does this mean that the large scale runs contain ad-hoc optimizations that are not used at lower resolution? What kind of fine-tuning was it needed? As stated, this bit gives no insight that could help other projects that are experiencing similar performance drops at large scale...

**Response:**

e) We appreciate the reviewer's interest in this point. The "fine-tuning of CPL and CICE" mentioned in the text primarily refers to process layout and communication optimizations at full machine scale. These optimizations are targeted for large-scale runs and do not affect experiments at smaller scales or lower resolutions, as already explained in Response 10.d.

**Comment 11**

**Reviewer comment:**

In section 4.4, figure 9 claims to show both ocean surface vorticity as well as sea ice concentration/deformation. However, I only see one colorbar, with a legend showings the units of vorticity. How should one read the sea ice concentration? The presence of green-ish and white-ish areas is also unclear. I suspect the white area is sea ice (but how does one deduce concentration/deformation?), and the green-ish area is just a tight overlap of areas with (small) positive an negative vorticity, which make blue and yellow pixel sit very near each other, causing a green effect. Still, some clarification may help the reader.

**Response:**

We appreciate the reviewer's valuable feedback. Figure 9 is plotted in the following order. The ocean surface relative vorticity is plotted first, and sea-ice variables are then overlaid. When the sea-ice concentration (SIC) is between 1% and 15%, the sea-ice concentration itself is displayed; in other regions where sea ice is present, the sea-ice total deformation rate is shown. In terms of color interpretation, white-to-blue shading represents sea-ice concentration (SIC), while red-to-white shading indicates the sea-ice total deformation rate. The greenish appearance mentioned by the reviewer mainly results from visual color mixing caused by the close spatial proximity of positive and negative vorticity and does not represent an additional physical quantity. In the revised manuscript, we have added colorbars and clarified the color interpretation in the figure legend and caption to avoid potential confusion for readers.

**Comment 12**

**Reviewer comment:**

In the "code and data availability" section at the end, the authors point to the ESCOMP org for the CESM code, and mention that the optimized code and the raw data is available upon request. I find this a bit underwhelming. I would prefer to see the optimized code as well as the input/run scripts (as well as any other input data) used for the experiments publicly available. They could easily create a zenodo (or similar) snapshot, as done in previous efforts by several climate modeling centers (as well as in previous works of some of the authors).

**Response:**

Thank you for your insightful comment regarding the "Code and Data Availability" section. We have added the relevant code and data links in the corresponding section of the revised manuscript.

**Comment 13**

**Reviewer comment:**

Finally, one comment on the keyword "automated" featured in the title. It is not clear to me how the optimization is "automated". Do the authors refer to the fact that the compiler does all the optimization work, once the proper pragma directives are inserted? If so, I find this a poor justification of the word "automated". Using OpenMP does not really qualify as "automated optimization". It is no more automated than the creation of the binary executable when running "make": it is just the compiler doing what it is designed to do. Moreover, the

authors explicitly say that they also needed precise fine-grain optimizations to squeeze out performance, which does not really ring with the "automated" tune of the title. To be clear, there is nothing wrong with having to manually modify/refactor selected areas of the code, and there is nothing wrong with relying on the compiler for vectorization and/or threading choices. But I would not call this an "automated optimization".

**Response:**

We thank the reviewer for raising the question regarding the term "automated" in the title. We understand the reviewer's concern about the concept of "automated optimization" and would like to clarify further. In this work, OpenMP directives indeed need to be manually inserted, which is a standard compiler parallelization practice. However, what we refer to as "automated optimization" primarily involves the ability to automatically generate worker thread code. In the revised manuscript, we have added clarifications in the revised manuscript. Please see line 370-373.